# DGMamba: Domain Generalization via Generalized State Space Model

## ABSTRACT

Domain generalization (DG) aims at solving distribution shift problems in various scenes. Existing approaches are based on Convolution Neural Networks (CNNs) or Vision Transformers (ViTs), which suffer from limited receptive fields or quadratic complexity issues. Mamba, as an emerging state space model (SSM), possesses superior linear complexity and global receptive fields. Despite this, it can hardly be applied to DG to address distribution shifts, due to the hidden state issues and inappropriate scan mechanisms. In this paper, we propose a novel framework for DG, named DGMamba, that excels in strong generalizability toward unseen domains and meanwhile has the advantages of global receptive fields, and efficient linear complexity. Our DGMamba compromises two core components: Hidden State Suppressing (HSS) and Semantic-aware Patch Refining (SPR). In particular, HSS is introduced to mitigate the influence of hidden states associated with domain-specific features during output prediction. SPR strives to encourage the model to concentrate more on objects rather than context, consisting of two designs: Prior-Free Scanning (PFS), and Domain Context Interchange (DCI). Concretely, PFS aims to shuffle the non-semantic patches within images, creating more flexible and effective sequences from images, and DCI is designed to regularize Mamba with the combination of mismatched non-semantic and semantic information by fusing patches among domains. Extensive experiments on four commonly used DG benchmarks demonstrate that the proposed DGMamba achieves remarkably superior results to state-of-the-art models. *The code will be made publicly available.*

## CCS CONCEPTS

• **Computing methodologies → Image representations**.

## KEYWORDS

Domain generalization, State space model, Mamba

## 1 INTRODUCTION

Humans are easily able to recognize images with domain distribution shifts (such as background changes [4] and various lighting conditions [53]) since the main semantic concepts are consistent. However, this is challenging for multimedia [19, 26, 41, 43, 47, 52, 65, 89, 91] and computer vision systems [3, 15, 27, 28, 51, 56]. One

**Unpublished working draft. Not for distribution.**

Permission to make digital or hard copies of all or part of this work for personal or classroom use is granted without fee provided that copies are not made or distributed for profit or commercial advantage and that copies bear this notice and the full citation on the first page. Copyrights for components of this work owned by others than the author(s) must be honored. Abstracting with credit is permitted. To copy otherwise, or republish, to post on servers or to redistribute to lists, requires prior specific permission and/or a fee. Request permissions from permissions@acm.org.

*ACM MM, 2024, Melbourne, Australia*

© 2024 Copyright held by the owner/author(s). Publication rights licensed to ACM.
ACM ISBN 978-x-xxxx-xxxx-x/YY/MM
https://doi.org/10.1145/nnnnnnn.nnnnnnn

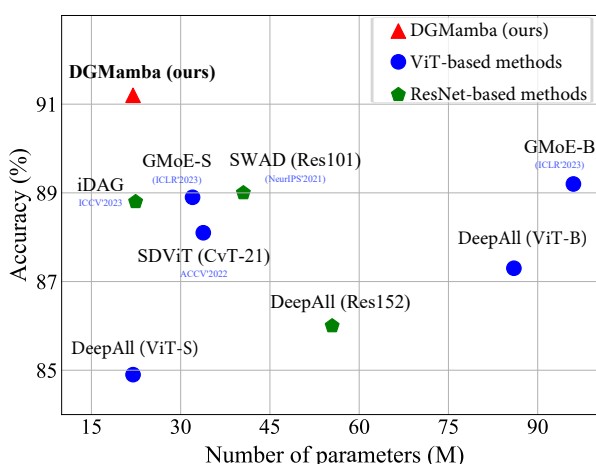

**Figure 1: Comparison of current CNN-based methods, ViT-based methods, and our proposed DGMamba on PACS of DG. Compared with these state-of-the-art (SOTA) methods, our proposed DGMamba achieves the best trade-off between the generalization performance (Accuracy) and computational complexity (Number of parameters).**

effective approach for eliminating distribution shifts is domain generalization (DG) [5, 20, 85], which attempts to encourage models to focus on semantic factors akin to humans and overlook non-essential features [17, 21, 38, 67, 69, 78, 83, 85].

A large amount of research in DG has concentrated on the design of special modules to acquire robust representation [62, 74, 78, 90], including domain alignment [16, 30, 45, 81], feature contrastive learning [32, 46], and style augmentation [8, 65, 75, 83, 88], *etc.* Nonetheless, prevailing DG methods heavily rely on CNNs as the backbone to extract latent features, only possessing local receptive fields. As a result, they tend to learn local details and overlook global information, impeding generalization and leading to less-desired performances on unseen domains. Recent advancements in DG [35, 49, 84] have shifted the backbone architecture from CNN [25] to ViT [11] due to its global receptive fields of self-attention layer [22, 35, 49, 50]. However, such attention layers in ViT introduce the challenge of quadratic complexity and lead to unacceptable computational inefficiency and memory overhead especially when models are very large, which makes it hard to deploy these DG methods in real-world applications.

Mamba, as an emerging state space model (SSM), possesses superior linear complexity and global receptive fields. It has been recently explored in language modeling and has a promising potential in computer vision. By employing input-dependent parameters in SSM, Mamba has exhibited promising performance in sequence data modeling and capturing long-range dependencies. In particular, VMamba [44] and Vision Mamba [92] propose to traverse the spatial domain and convert any non-causal visual image into order patch sequences. However, such SSM-based models inevitably

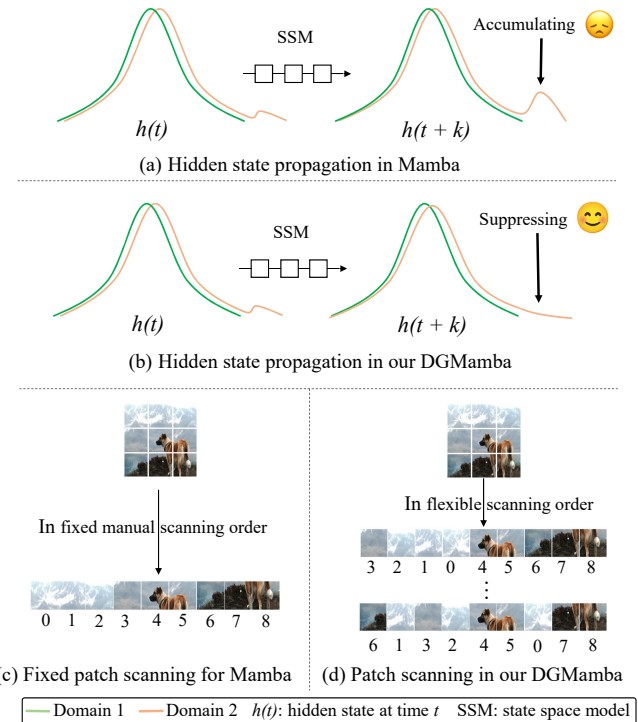

**Figure 2: (a) When directly adapting Mamba to DG, domain-specific information captured by hidden states may be accumulated or even amplified during the hidden state propagation, impeding the generalizability. (b) In contrast, Hidden State Suppressing (HSS) is introduced in our DGMamba to alleviate the adverse effect of domain-specific information in hidden states. (c) Simple and fixed scanning strategies for Mamba may result in unexpected domain-specific information within the generated sequence data when scanning 2D images into 1D sequences, thereby undermining the ability of Mamba to address distribution shifts. (d) In contrast, the proposed Prior-Free Scanning in DGMamba endeavors to break the prior bias introduced by the fixed manual scanning, offering more meaningful sequence data.**

exhibit performance degradation in unseen domains due to the lack of consideration of domain shifts and tailored designs, and there still exist generalization performance gaps compared with the state-of-the-art DG methods, *e.g.,* iDAG [30] on the PACS dataset (87.7% vs. 88.8%). Therefore, how to improve the generalizability of Mamba-based models and investigate what hinders Mamba from addressing distribution shifts for DG is a very critical problem.

In this paper, our goal is to enhance the generalizability of Mamba-like models toward unseen domains. Our motivations mainly lie in two aspects. *Firstly*, we observe that hidden states, as an essential part of Mamba, play an important role in modeling long-range correlations by recording the historical information in sequence data, facilitating global receptive fields. However, when dealing with unseen images containing diverse domain-specific information from varying domains, such hidden states may yield undesirable effects. The domain-specific information could potentially be accumulated or even amplified in hidden states during propagation,

as indicated in Figure 2 (a), thereby degrading the generalization performance. *Secondly*, how to effectively scan 2D images into 1D sequence data that is suitable for Mamba in DG is still an open problem since the pixels or patches of images do not exhibit the necessary causal relations existed in sequence data. Although recent works [37, 44, 92] have explored various scanning strategies for vision tasks, such simple 1D traverse strategies may result in unexpected domain-specific information within the generated sequence data (Figure 2(c)), thereby undermining the ability of Mamba to address distribution shifts. Besides, these fixed scanning strategies largely overlook domain-agnostic scanning and are highly sensitive to various varying scenarios, making it difficult to apply to DG.

Motivated by the above facts, we propose DGMamba, a novel State Space Model-based framework for domain generalization that excels in strong generalizability toward unseen domains and meanwhile has the advantages of global receptive fields, and efficient linear complexity. DGMamba comprises **two** key modules, Hidden State Suppressing (HSS) and Semantic-aware Patch Refining (SPR). *Firstly*, HSS is presented to eliminate the detrimental effect of non-semantic information contained in hidden states by selectively suppressing the corresponding hidden states during output prediction. By reducing non-semantic information in SSM layers, DGMamba learns domain invariant features. *Secondly*, SPR is introduced to encourage the model to pay more attention to objects rather than context, consisting of two key designs: Prior-Free Scanning (PFS), and Domain Context Interchange (DCI). Specifically, PFS is designed to shuffle the context patches within images that contribute less to the label prediction. It provides an effective 2D scanning mechanism to traverse 2D images into 1D sequence data. As a result, PFS possesses the ability to shift the model's attention from the context to the object. Besides, to alleviate the influence of diverse context information and local texture details across different domains, DCI replaces the context patches of images with those from different domains. The proposed DCI brings in local texture noise and regularizes the model on the combination of mismatched context and object. By leveraging both linear complexity and heterogeneous context tokens, DCI learns more robust representation efficiently. Aggregating all these contributions into one architecture, our proposed DGMamba achieves strong results on four DG benchmarks. As shown in Figure 1, compared with previous CNN-based and ViT-based methods, our DGMamba achieves the best trade-off between accuracy and parameters. In summary, we make the following contributions:

- We propose DGMamba, a novel State Space Model-based framework for domain generalization that excels in strong generalizability toward unseen domains and meanwhile has the advantages of global receptive fields and efficient linear complexity. To the best of our knowledge, this is the first work that studies the generalizability of the SSM-based model (Mamba) in domain generalization.
- We present Hidden State Suppressing (HSS) and Semantic-aware Patch Refining (SPR) to improve the generalizability of the SSM-based model. Concretely, HSS is introduced to mitigate the detrimental influence rising from the hidden states, reducing the gap between hidden states across domains. SPR, comprising two modules, namely PFS and DCI,

is designed to augment the context environments to shift the model's attention to the object.

- Extensive experiments with analyses on widely used benchmarks in DG show that our presented DGMamba achieves state-of-the-art generalization performance, showcasing its effectiveness and superiority in boosting the generalizability toward unseen domains.

## 2 RELATED WORK

**CNN-based models in DG** employ convolution neural networks, *e.g.*, Alexnet [33] and ResNet [25], to extract stronger representations. These approaches specially designed submodules to regularize the acquired features. The most intuitive idea for DG is to minimize the empirical source risk [1, 29, 70, 80]. Domain alignment [16, 39, 63, 79, 81] constrained models to convey little domain characteristics by an extra domain discriminative network. Feature disentanglement methods [40, 64, 65, 73] aim to disentangle the features to acquire task-specific information. Another significant avenue is to augment the source data [55, 61, 86, 86, 88], thereby providing models with more samples with diverse styles. Contrastive learning [31, 31, 31, 32, 46, 71] employed the contrastive loss function on features to reduce the gap of representation distributions in one category. Ensemble learning [7, 9, 32] utilized stochastic weight average to find a flatter minimum in loss spaces to enhance generalizability. Approaches based on meta-learning [2, 12, 13, 82] attempt to address distribution shifts during the training phase, enabling models to learn to tackle domain shifts. Despite their remarkable progress in DG and the linear complexity, the lack of global receptive fields hinders further developments of CNN-based models for boosting generalization performance [35, 44].

**ViT-based models in DG** leverage ViT [11] as the backbone to learn high-quality representations [49, 72], harnessing the merit of global receptive fields inherent in ViT. Discovering that the architecture of ViT aligns better with the invariant correlations than CNN [35], GMoE [35] utilized a generalizable mixture of experts to capture diverse attributes with different experts effectively. SD-ViT [57] attempted to tackle the overfitting in source domains by guaranteeing better prediction results from the tokens in intermediate ViT layers. TFS-ViT [49] augmented the feature styles by token replacement. Despite the inherent advantages in global receptive fields, ViT-based models suffer from the quadratic complexity with respect to the image resolution rising from the attention mechanism, leading to extra overhead of computation and memory.

**Mamba** has been widely explored in vision task [24, 37, 42, 44, 66, 68, 77, 92] to integrate both excellence of global receptive fields and linear complexity. VMamba [44] and Vim [92] proposed visual state space models to deploy Mamba for vision tasks. PCM [77] employed Mamba in point cloud analysis by introducing merged point prompts. Unfortunately, rare research has been conducted on the generalization performance of Mamba in vision tasks. To our knowledge, this is the first time that the SSM-based model Mamba has boosted the model generalization performance.

## 3 METHOD

In this section, we begin by introducing the concepts related to Mamba [18], *i.e.*, the State Space Model (SSM), and the selective

scan mechanism. Based on this, we propose DGMamba to enhance the model generalization performance. As shown in Figure 3, DG-Mamba includes two core modules: Hidden State Suppressing (HSS), and Semantic-aware Patch Refining (SPR). HSS serves to suppress the domain-specific information conveyed in the hidden states by reducing the corresponding weighting factors. SPR encourages the model to focus more on the *object* instead of the *context*. SPR includes two core elements: Prior-Free Scanning (PFS), and Domain Context Interchange (DCI). PFS augments the range of the scanning mechanism of Mamba by randomly shuffling the non-semantic patches within images, and DCI attempts to distort images by substituting context patches with those in other domains.

### 3.1 Preliminaries

**State Space Model (SSM).** Derived from linear time-invariant systems, SSM-based models endeavor to establish a correlation between signals $x(t) \in \mathbb{R}^L$ and the resultant response $y(t) \in \mathbb{R}^L$ via the hidden state $h(t) \in \mathbb{R}^N$. Mathematically, these models can be represented as linear ordinary differential equations (ODEs), as denoted by Eq. (1):

$$h'(t) = Ah(t) + Bx(t), \ y(t) = Ch(t), \tag{1}$$

where the parameters encompass $A \in \mathbb{R}^{N \times N}$, $B, C \in \mathbb{R}^N$, with $N$ denoting the state size.

When applied to deep learning, time-continuous SSMs require adjustment through discretization to align with the input sample rate. Based on the discretization methodology in [23], the ODE depicted in Eq. (1) can be discretized as following:

$$h_t = \bar{A}h_{t-1} + \bar{B}x_t, \quad y_t = Ch_t,$$
$$\bar{A} = e^{\Delta A}, \quad \bar{B} = (\Delta A)^{-1} \left( e^{\Delta A} - I \right) \cdot \Delta B, \tag{2}$$

where $\Delta$ represents the sample parameter of the inputs, facilitating the discretization process.

**2D Selective Scan Mechanism.** In addition to the challenge posed by the inconsistency between the time-continues system and discretized signals, the characteristics of multi-media signals, such as images and videos, mismatch the architecture of the SSM-based models, which are designed to capture information within temporal signals or sequence data. As a different modal from language, images contain ample spatial information, including local texture and global shape, which may not exhibit causal correlations in sequence data. To tackle this problem, the selective scan mechanism becomes imperative. Existing methods [37, 44, 92] tend to scan images into sequence data in a fixed manner. For instance, in VMamba [44], images are flattened into two patch sequences along row and column, respectively. VMamba models these two sequences by scanning forward and backward, respectively.

**Shortcomings of Mamba for DG.** However, the naive Mamba network encounters challenges when confronted with distribution shifts in DG, achieving a generalization performance of 87.7% on PACS dataset, inferior to the existing DG method iDAG [30] (88.8%). In the following section, we will delve into the reason behind this performance gap and propose effective strategies to assist Mamba in tackling distribution shifts for DG.

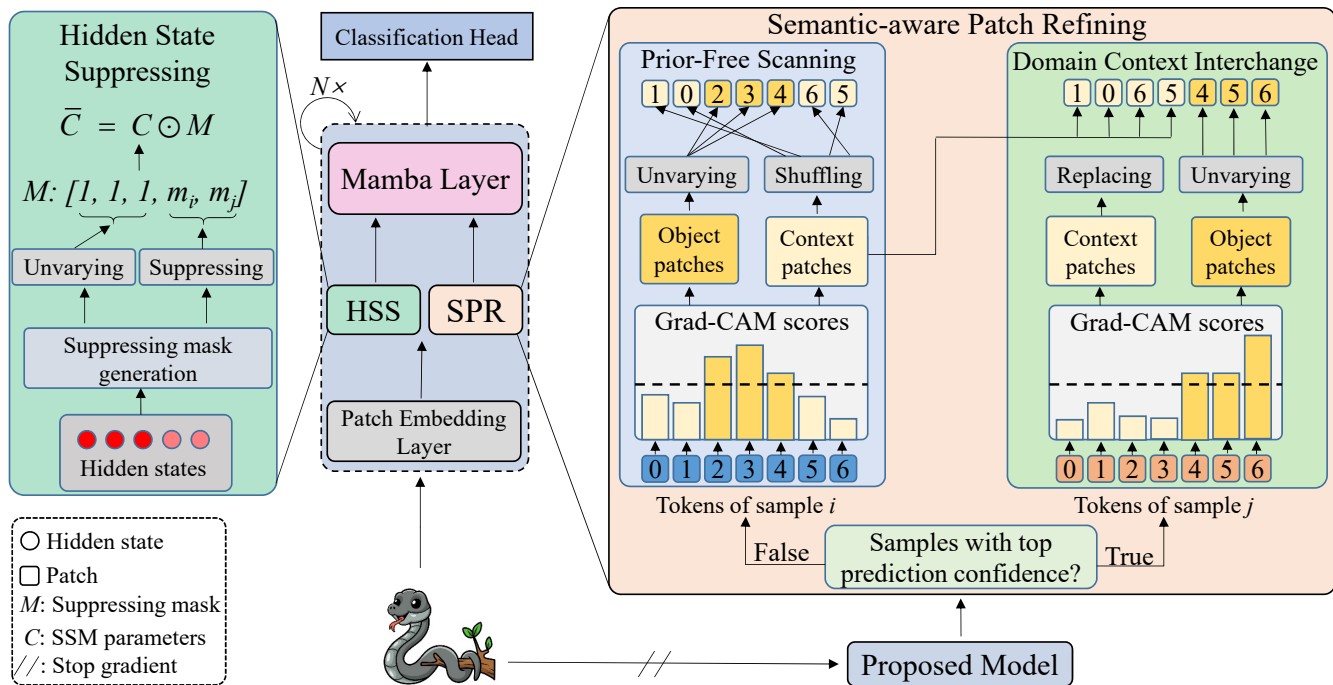

**Figure 3: The framework of our DGMamba. Before passing the patches into Mamba layers, the proposed Semantic-aware Patch Refining (SPR) is employed. Concretely, for the samples not in the top percentage of prediction confidence, we apply Prior-Free Scanning to randomly shuffle the background patches that exhibit low Grad-CAM scores, providing DGMamba with a more flexible and effective 2D scanning mechanism. For the remaining samples, we substitute their background patches with those from diverse domains, introducing texture noise and context confusion to avoid overfitting. In addition, the Hidden State Suppressing (HSS) is introduced to reduce the importance of hidden states that comprise domain-specific information.**

## 3.2 Hidden State Suppressing

Domain-specific information poses a great challenge for deep learning models, including the SSM-based models [23, 44, 77]. In these SSM-based models, the hidden states $h(t)$ play an essential role in capturing long-range correlations by propagating historical information along the sequence data. It accumulates and propagates information from previous time steps, allowing the model to remember past states and information and thereby endowing the model with global receptive fields.

Despite the significant role of hidden states in Mamba, the working mechanism of hidden states could bring about negative influences when facing distribution shifts. As shown in Figure 2, when confronted with images from diverse domains, domain-specific information could also be accumulated or even be increased in hidden states, impeding model generalizability. To mitigate the influence of accumulated domain-specific information in hidden states when predicting $y$, we propose to suppress the corresponding parts in the hidden states $h(t)$, which may carry domain-specific information.

In order to suppress the unexpected domain-specific information conveyed in the hidden states, the initial task is to recognize the hidden states that hold these adverse factors. According to the propagation rule of hidden states in Eq. (2), $\Delta A$ (showing a positive correlation with $\bar{A}$) dominates the transition of hidden states. While the hidden states $h_t$ serve a pivotal role when predicting the output $y_t$. Thus, hidden states that show stronger correlations with

the genuine label should be preserved more prominently during the propagation of hidden states. Consequently, they necessitate larger propagation coefficients in $\bar{A}$, while less associated hidden states require comparatively smaller coefficients in $\bar{A}$. As a result, the value of $\Delta A$ is utilized to determine which hidden states will undergo suppression. Mathematically, the proposed Hidden State Suppressing (HSS) strategy can be formulated as follows:

$$y_t = \bar{C}h_t, \quad \bar{C} = C \odot M,$$
$$M = (\Delta A > \alpha) + (1 - (\Delta A > \alpha)) \odot \Delta A, \tag{3}$$

where $\alpha \in [0, 0.5]$ represents the threshold for determining whether the hidden states should be suppressed. In this way, the hidden states, whose coefficient parameters $\Delta A <= \alpha$ will be suppressed by $\Delta A$, while the remaining hidden states remain the same.

## 3.3 Semantic-aware Patch Refining

In addition to the proposed HSS in eliminating the adverse effect of domain-specific information, enforcing the model to pay more attention to the *object* rather than the *context* can also be an effective way to facilitate the generalization performance.

From the perspective of domain-invariant learning, *context* and *object* are two basic elements. The *object* corresponds to the foreground, which contributes most to the classification results, remaining stationary in diverse scenarios. The *context* is related to domain-specific information, such as background and image style,

**Table 1: Results on PACS with our DGMamba.**

| Method | Params. | Art | Cartoon | Photo | Sketch | Avg.(↑) |
|--------|---------|-----|---------|-------|--------|---------|
| | | ResNet50 | | | | |
| VREx [34] | 23M | 86.0 | 79.1 | 96.9 | 77.7 | 84.9 |
| MTL [6] | 23M | 87.5 | 77.1 | 96.4 | 77.3 | 84.6 |
| Mixstyle [87] | 23M | 86.8 | 79.0 | 96.6 | 78.5 | 85.2 |
| SagNet [48] | 23M | 87.4 | 80.7 | 97.1 | 80.0 | 86.3 |
| ARM [76] | 23M | 86.8 | 76.8 | 97.4 | 79.3 | 85.1 |
| SWAD [7] | 23M | 89.3 | 83.4 | 97.3 | 82.5 | 88.1 |
| PCL [71] | 23M | 90.2 | 83.9 | 98.1 | 82.6 | 88.7 |
| SAGM [62] | 23M | 87.4 | 80.2 | 98.0 | 80.8 | 86.6 |
| iDAG [30] | 23M | 90.8 | 83.7 | 98.0 | 82.7 | 88.8 |
| GMDG [58] | 23M | 84.7 | 81.7 | 97.5 | 80.5 | 85.6 |
| | | DeiT-S | | | | |
| SDViT [57] | 22M | 87.6 | 82.4 | 98.0 | 77.2 | 86.3 |
| GMoE [35] | 34M | 89.4 | 83.9 | **99.1** | 74.5 | 86.7 |
| | | VMamba-T | | | | |
| DGMamba (ours) | 22M | **91.3** | **87.0** | 99.0 | **87.3** | **91.2** |

**Table 2: Results on VLCS with our DGMamba.**

| Method | Params. | Caltech | LabelMe | SUN | PASCAL | Avg.(↑) |
|--------|---------|---------|---------|-----|--------|---------|
| | | ResNet50 | | | | |
| VREx [34] | 23M | 98.4 | 64.4 | 74.1 | 76.2 | 78.3 |
| MTL [6] | 23M | 97.8 | 64.3 | 71.5 | 75.3 | 77.2 |
| Mixstyle [87] | 23M | 98.6 | 64.5 | 72.6 | 75.7 | 77.9 |
| SagNet [48] | 23M | 97.9 | 64.5 | 71.4 | 77.5 | 77.8 |
| ARM [76] | 23M | 98.7 | 63.6 | 71.3 | 76.7 | 77.6 |
| SWAD [7] | 23M | 98.8 | 63.3 | 75.3 | 79.2 | 79.1 |
| PCL [71] | 23M | 99.0 | 63.6 | 73.8 | 75.6 | 78.0 |
| SAGM [62] | 23M | **99.0** | 65.2 | 75.1 | 80.7 | 80.0 |
| iDAG [30] | 23M | 98.1 | 62.7 | 69.9 | 77.1 | 76.9 |
| GMDG [58] | 23M | 98.3 | **65.9** | 73.4 | 79.3 | 79.2 |
| | | DeiT-S | | | | |
| SDViT [57] | 22M | 96.8 | 64.2 | 76.2 | 78.5 | 78.9 |
| GMoE [35] | 34M | 96.9 | 63.2 | 72.3 | 79.5 | 78.0 |
| | | VMamba-T | | | | |
| DGMamba (ours) | 22M | 98.9 | 64.3 | **79.2** | **80.8** | **80.8** |

which varies dramatically across domains. Therefore, directing the model's focus toward the *object* could assist in mitigating the domain-specific information. As a result, we propose Semantic-aware Patch Refining (SPR) to assist the model in better focusing on the *object*. SPR consists of two core modules: Prior-Free Scanning (PFS) and Domain Context Interchange (DCI). SPR is devoted to constructing a sufficient and random context environment, thereby breaking the adverse effect of the domain-specific information implied by the *context* in the input and enhancing generalizability.

*Prior-Free Scanning.* Although SSM-based models [44, 92] have demonstrated excellent performance in vision tasks, a diverse and random context environment is still essential to deploy Mamba in DG. This conclusion manifests that an effective scanning mechanism is still required to tackle the challenge posed by the non-causal correlations between image pixels or patches. Suitable scanning mechanisms [44, 77, 92] should possess the ability to break unexpected spurious correlations caused by the manually created sequences of images. Nevertheless, existing SSM-based methods [44, 77, 92] are limited to scanning images into patches in a fixed unfolding approach, as indicated in Figure 2(c). These subjective traverse strategies could result in domain-specific information in the generated sequence, making it difficult for these models to address the distribution shifts in DG.

To break the spurious correlations between patches and provide an effective scanning mechanism for DG tasks, we propose Prior-Free Scanning (PFS) to address the direction-sensitive issue in Mamba. As depicted in Figure 3, PFS attempts to randomly shuffle the *context* patches that may contribute to domain-specific information in the unfolded sequence, while keeping the *object* patches unchanged. In particular, for the representation $z = z_c + z_o \in \mathbb{R}^{H \times W \times C}$, where $z_c$ and $z_o$ represent the *context* information and *object* information, respectively, the shuffled representation $z_{pfs}$ after the PFS strategy can be formulated as follows:

$$z_{pfs} = z_c^s + z_o, \quad z_c^s = Shuffle(z_c), \quad (4)$$

where $z_c^s$ denotes the shuffled *context* information by employing the $Shuffle(\cdot)$ function in the spatial dimension. This operation could

provide Mamba with sequence data exhibiting flexible scanning direction by generating *context* disturbance or noise while keeping the consistent *object* information. As a result, it mitigates the domain-specific information brought by the manual fixed scanning strategy and breaks the spurious correlations.

*Domain Context Interchange.* The proposed PFS facilitates the model to pay more attention to the *object* instead of the *context* within images. However, the *context* information is heterogeneous across varying domains in DG. The *context* patch shuffling in PFS is limited in the given scenarios, inadequate to provide sufficient diverse *context* information for removing the domain-specific information. Besides, the heterogeneous *context* patches from different domains not only exhibit diverse *context* information but also encompass distinct local texture characteristics.

To sufficiently tackle the adverse influence of heterogeneous *context* and diverse local texture details, we propose to create ample *context* scenarios and introduce local texture noise by Domain Context Interchange (DCI). As shown in the Domain Context Interchange module in Figure 3, DCI substitutes the image *context* patches with those from different domains. This operation regularizes the model on the counterfacture samples [8, 69], *i.e.*, the combination of semantic information in one domain and non-semantic features from different domains. This strategy further forces models to focus on the generalizable features while discarding the textual details or other domain-specific features.

In implementation, DCI only performs on samples with high confidence in the classification results, while other samples remain unchanged. As the samples exhibiting high classification confidence may result in overfitting in their scenarios, replacing their *context* patches with those from different domains could introduce challenging *context* noise to generate heterogeneous *context* and local texture noise. While remaining samples may struggle to recognize the *object* from the *context*, conducting DCI on them could further increase the difficulty in learning generalizable presentations. Specifically, we only apply DCI to the top 20% of batch samples according to the classification confidence based on the negative cross-entropy loss. These samples with high negative cross-entropy

**Table 3: Results on OfficeHome with our DGMamba.**

| Method | Params. | Art | Clipart | Product | Real | Avg.(↑) |
|--------|---------|-----|---------|---------|------|---------|
| ResNet50 | | | | | | |
| VREx [34] | 23M | 60.7 | 53.0 | 75.3 | 76.6 | 66.4 |
| RSC [29] | 23M | 60.7 | 51.4 | 74.8 | 75.1 | 65.5 |
| MTL [6] | 23M | 61.5 | 52.4 | 74.9 | 76.8 | 66.4 |
| Mixstyle [87] | 23M | 51.1 | 53.2 | 68.2 | 69.2 | 60.4 |
| SagNet [48] | 23M | 63.4 | 54.8 | 75.8 | 78.3 | 68.1 |
| ARM [76] | 23M | 58.9 | 51.0 | 74.1 | 75.2 | 64.8 |
| SWAD [7] | 23M | 66.1 | 57.7 | 78.4 | 80.2 | 70.6 |
| PCL [71] | 23M | 67.3 | 59.9 | 78.7 | 80.7 | 71.6 |
| SAGM [62] | 23M | 65.4 | 57.0 | 78.0 | 80.0 | 70.1 |
| iDAG [30] | 23M | 68.2 | 57.9 | 79.7 | 81.4 | 71.8 |
| GMDG [58] | 23M | 68.9 | 56.2 | 79.9 | 82.0 | 70.7 |
| DeiT-S | | | | | | |
| SDViT [57] | 22M | 68.3 | 56.3 | 79.5 | 81.8 | 71.5 |
| GMoE [35] | 34M | 69.3 | 58.0 | 79.8 | 82.6 | 72.4 |
| VMamba-T | | | | | | |
| DGMamba (ours) | 22M | **76.2** | **61.8** | **83.9** | **86.1** | **77.0** |

**Table 4: Results on TerraIncognita with our DGMamba.**

| Method | Params. | L100 | L38 | L43 | L46 | Avg.(↑) |
|--------|---------|------|-----|-----|-----|---------|
| ResNet50 | | | | | | |
| VREx [34] | 23M | 48.2 | 41.7 | 56.8 | 38.7 | 46.4 |
| RSC [29] | 23M | 50.2 | 39.2 | 56.3 | 40.8 | 46.6 |
| MTL [6] | 23M | 49.3 | 39.6 | 55.6 | 37.8 | 45.6 |
| Mixstyle [87] | 23M | 54.3 | 34.1 | 55.9 | 31.7 | 44.0 |
| SagNet [48] | 23M | 53.0 | 43.0 | 57.9 | 40.4 | 48.6 |
| ARM [76] | 23M | 49.3 | 38.3 | 55.8 | 38.7 | 45.5 |
| SWAD [7] | 23M | 55.4 | 44.9 | 59.7 | 39.9 | 50.0 |
| PCL [71] | 23M | 58.7 | 46.3 | 60.0 | 43.6 | 52.1 |
| SAGM [62] | 23M | 54.8 | 41.4 | 57.7 | 41.3 | 48.8 |
| iDAG [30] | 23M | 58.7 | 35.1 | 57.5 | 33.0 | 46.1 |
| GMDG [58] | 23M | 59.8 | 45.3 | 57.1 | 38.2 | 50.1 |
| DeiT-S | | | | | | |
| SDViT [57] | 22M | 55.9 | 31.7 | 52.2 | 37.4 | 44.3 |
| GMoE [35] | 34M | 59.2 | 34.0 | 50.7 | 38.5 | 45.6 |
| VMamba-T | | | | | | |
| DGMamba (ours) | 22M | **62.7** | **48.3** | **61.1** | **46.4** | **54.6** |

loss are easy samples for the model, leading to inflated confidence in their classification, which may, in turn, result in overfitting issues. **Context Patch Identifying.** To distinguish the *context* and *object* patches, we take advantage of Grad-CAM [54] as the metric to measure the contributions of different regions of images. As the regions containing the *object* would activate the Grad-CAM greatly, while the patches exhibiting the *context* possess a low value in the Grad-CAM, we split the image patches into *context* and *object* according to their values in the activation map generated by Grad-CAM. Specifically, the patches with the smallest 25% Grad-CAM values are determined as the *context* information $z_c$, while the remaining are used as *object* information $z_o$.

## 4 EXPERIMENTS

**Dataset.** Following standard protocol in DG [7, 20], we evaluate the effectiveness of our proposed DGMamba and compare it with state-of-the-art methods in DG on four commonly used benchmarks: (1) PACS [36] includes 9991 images categorized into 7 classes exhibiting 4 styles. (2) VLCS [14] involves four datasets, totaling 10729 images distributed in 5 categories. (3) OfficeHome [60] comprises 15588 images in 65 classes from 4 datasets. (4) TerraIncognita [4] encompasses 24330 photographs of 10 kinds of animals taken at 4 diverse locations.

**Implementation Details.** Our proposed model employs Mamba [18] as the backbone, which is pretrained on ImageNet [10]. Following VMamba [44], the network comprises 4 blocks, each consisting of 2, 2, 4, and 2 Mamba layers, respectively. Down-sampling is incorporated before each block. Additionally, bidirectional Mamba is utilized to enable each patch to gather information from any other patches. Following the training configuration in existing DG approaches [7, 20, 35] and considering the limitations of the GPU, the model undergoes training for 10000 iterations, with a batch size of 16 for each source domain. For optimization, we employ the AdamW optimizer with betas of (0.9, 0.999) and a momentum of 0.9. We incorporate a cosine decay learning rate scheduler. The initial learning rate is searched in [3e-4, 4.5e-4].

### 4.1 Main Results

**Results on PACS Dataset.** Table 1 reports the generalization performance on PACS, indicating the best generalization performance of our DGMamba across almost all these domains. Notably, DG-Mamba surpasses the SOTA methods by 2.7% in average generalization performance. Besides, on the hard-to-transfer domains, such as 'Cartoon' and 'Sketch,' our approach beats the second-best method by 3.9% and 5.6%, respectively, demonstrating the effectiveness of our DGMamba in enhancing generalization capacity.

**Results on VLCS Dataset.** As shown in Table 2, our DGMamba demonstrates superior average generalization performance compared to SOTA methods. In addition, our method consistently ranks among the top three performers in three out of the four scenarios, indicating its efficacy in improving the model's generalizability.

**Results on OfficeHome Dataset.** The results on OfficeHome are shown in Table 3. The proposed DGMamba has achieved a significant enhancement in generalization performance in all scenarios. Remarkably, DGMamba outperforms the SOTA method by 6.4% in average generalization performance. These findings showcase the superiority of DGMamba in acquiring robust representations.

**Results on TerraIncognita Dataset.** We provide the experiment results on TerraIncognita in Table 4. Our proposed DGMamba attains the best generalization performance across all environments. Notably, our proposed approach achieves a 4.8% gain over the SOTA method in average generalization performance. These outcomes highlight DGMamba's outstanding ability to obtain domain-invariant representations.

### 4.2 Ablation Study and Analysis

**Effectiveness of Each Component.** We conduct an ablation study on PACS to disclose the contributions of our proposed modules on the generalization performance. As indicated in Table 5, the proposed HSS, PFS, and DCI consistently facilitate the enhancement of generalizability across almost all scenarios, demonstrating their effectiveness in capturing genuine correlations and removing domain-specific information. Specifically, in the hardest-to-transfer

**Table 5: Ablation study on PACS with our proposed modules.**

| Method | Art | Cartoon | Photo | Sketch | Avg.(↑) |
|---|---|---|---|---|---|
| VMamba [44] | 88.2 | 86.2 | 98.4 | 84.9 | 89.4 |
| w/ HSS | 90.4 | 86.8 | 98.8 | 87.1 | 90.8 |
| w/ PFS | 91.8 | 85.9 | 98.7 | 85.4 | 90.5 |
| w/ DCI | **91.8** | 85.8 | 99.0 | 85.3 | 90.4 |
| DGMamba | 91.3 | **87.0** | **99.0** | **87.3** | **91.2** |

**Table 6: Effectiveness of our proposed SPR when inserting at different state space blocks of DGMamba on PACS.**

| Block | Art | Cartoon | Photo | Sketch | Avg.(↑) |
|---|---|---|---|---|---|
| None | 88.2 | 86.2 | 98.4 | 84.9 | 89.4 |
| Block 3 | 91.7 | **86.8** | 98.9 | 82.7 | 90.0 |
| Block 2 | 91.7 | 86.5 | 98.9 | 83.9 | 90.2 |
| Block 1 | **92.3** | 86.3 | **99.1** | **85.4** | **90.8** |

**Table 7: Comparison of the proposed Hidden State Suppressing (HSS) with Hidden State Masking (HSM) on PACS.**

| Threshold | 0 | 0.15 | 0.2 | 0.25 |
|---|---|---|---|---|
| HSM | 89.4 | 90.2 | 90.3 | 90.1 |
| HSS | 89.4 | 90.1 | 90.5 | 90.8 |

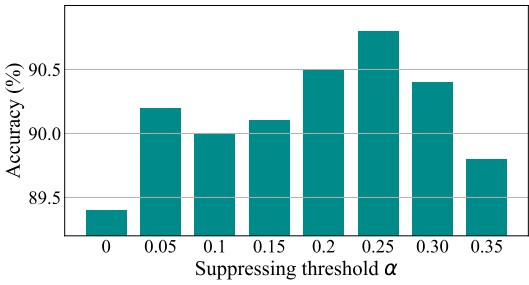

**Figure 4: Effect of $\alpha$ in the proposed HSS on PACS.**

**Table 8: Comparison of the computational efficiency of existing SOTA methods in DG and our DGMamba on PACS. Tested with an image size of 224× 224 on one NVIDIA Tesla V100.**

| Model | Backbone | Params | GFlops | Time | Acc (%) |
|---|---|---|---|---|---|
| iDAG [30] | ResNet50 | 23M | 8G | 94ms | 88.8 |
| iDAG [30] | ResNet101 | 41M | 15G | 495ms | 89.2 |
| GMoE-S [35] | DeiT-S | 34M | 5G | 136ms | 88.1 |
| GMoE-B [35] | DeiT-B | 133M | 19G | 361ms | 89.2 |
| VMamba [44] | VMamba-T | 22M | 5G | 225ms | 89.4 |
| DGMamba (ours) | VMamba-T | 22M | 5G | 233ms | 91.2 |

domain where VMamba behaves poorly, *i.e.*, 'Sketch', HSS shows the greatest enhancement, with an increase of 2.6%. This underscores the superiority of HSS in mitigating the domain-specific information carried in hidden states. The proposed PFS could also enhance the performance in the remaining domains, underscoring that offering a more effective scanning strategy and directing the model's focus on the object are beneficial to improving generalizability. In addition, the heterogeneous context and texture noise introduced by DCI can also promote the model to shift attention to domain invariant features. Moreover, the combination of these proposed modules reaches the highest performance, indicating the necessity of these modules in yielding optimal performance enhancement.

**SPR at Different Layers.** We insert the SPR module at different layers of DGMamba and showcase diverse performance gains in Table 6. The shallow layers contain more spurious correlations, and thereby, should benefit more from our SPR module than the deeper layers, which hold more correlations with the prediction results. This inference is also supported by the generalization performance in Table 6. The highest gains in generalization performance have been achieved when inserting the SPR module before Block 1.

**Effect of $\alpha$ in HSS.** The threshold $\alpha$ in HSS controls the number of hidden states to be suppressed. Intuitively, more suppressed hidden states would eliminate more domain-specific information, thereby enhancing generalization performance. Figure 4 reports the generalization performance of our HSS by varying $\alpha$, indicating that the best performance can be obtained by HSS when the threshold $\alpha$ is set to 0.25. This observation demonstrates the effectiveness of HSS in mitigating the detrimental effect of domain-specific features on generalization. However, it is noteworthy that a relatively large $\alpha$ would also mitigate the positive influences of hidden states associated with the valuable features for predictions, consequently degrading the generalization performance.

**Hidden State Suppressing (HSS) *vs*. Hidden State Masking (HSM).** In practice, we also experiment with a hidden state masking strategy to eliminate the adverse effect of domain-specific information. HSM

just attempts to discard the hidden states associated with domain-specific information, assuming these hidden states contribute little to the predictions. However, as indicated in Table 7, although exhibiting the ability to enhance generalization performance, HSM is less competitive than our HSS. The result demonstrates that these hidden states deserve to be suppressed and may also convey essential semantic information to capture long-range correlations.

**Computation Efficiency.** To assess the computational efficiency of our proposed DGMamba, we conduct experiments on PACS to compare it with SOTA methods from different perspectives, including model parameters, float-point-operations per second (Flops), inference time and their generalization performance on PACS. It is worth noting that iDAG [30] requires multiple samples in the training phase, thus we create a tensor with batch size 2 to evaluate its GFlops. For the remaining methods, the dimension of batch size for evaluating Flops is set to 1. The inference time is averaged over 100 experiments. As indicated in Table 8, while our proposed DGMamba exhibits relatively fewer parameters and GFlops than existing SOTA methods based on CNN or ViT, it still achieves the highest generalization performance.

**Feature Visualization.** To visually demonstrate the impact of our proposed DGMamba, we adopt t-SNE embeddings [59] to investigate the feature characteristics. Concretely, we conduct experiments on PACS with 'Photo' as the target domain. Figure 5 depicts the feature visualizations based on the CNN-based method iDAG [30], ViT-based method GMoE [35], VMamba [44], and our proposed DGMamba, respectively. Remarkably, DGMamba acquires superior representations, exhibiting enhanced intra-class compactness and inter-class discrimination, especially in comparison with iDAG and GMoE. Besides, the enhancement of DGMamba relative to VMamba is also obvious. Firstly, the distinction between features of 'dog' (blue) and 'horse' (purple) in DGMamba is more pronounced. Secondly, the features within the same category in DGMamba manifest increased compactness. These findings confirm the superiority of DGMamba in boosting model generalization capacity.

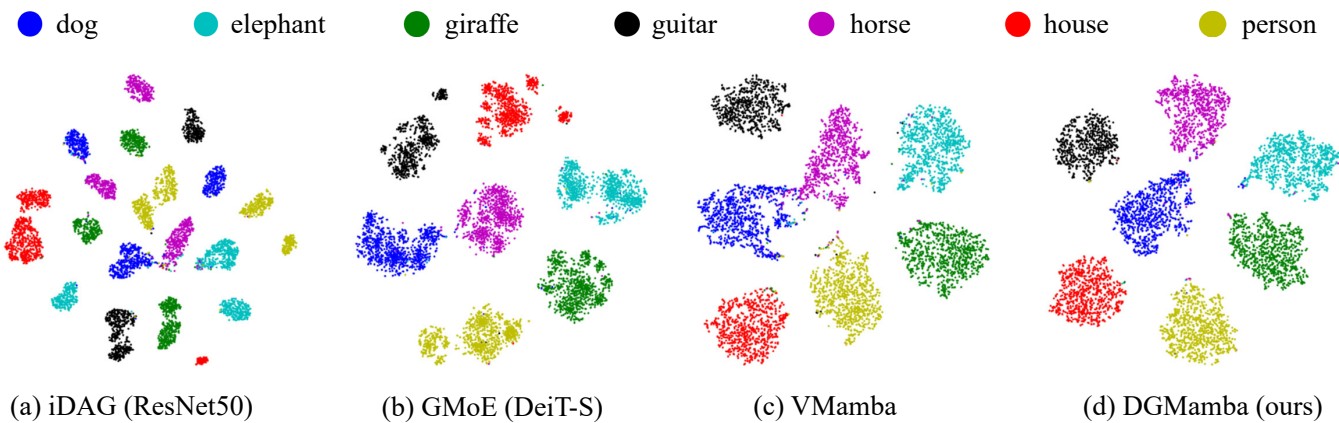

Figure 5: Visualizations with t-SNE embeddings [59] illustrating various classes' representations produced by (a) iDAG [30], (b) GMoE [35], (c) VMamba [44], and (d) DGMamba (ours), respectively. DGMamba demonstrates the superior clustering effect.

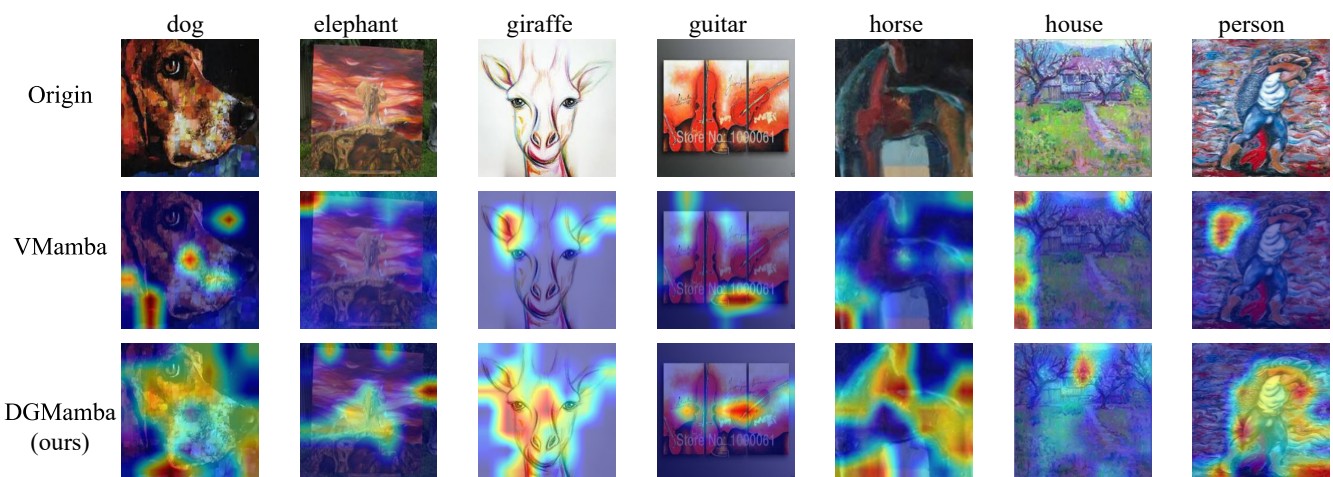

Figure 6: Visualization for the activation maps of the last state space layer on PACS with 'Art' as the target domain. For each sample, the first row represents the original image, the middle row is the activation map generated by VMamba without any domain generalization techniques, and the last row denotes the activation map captured by our proposed DGMamba.

**Activation Map Visualization.** To further visually verify the outstanding prediction mechanism of our proposed DGMamba, we provide visualizations for the activation maps of the last state space layer with Grad-CAM techniques [54]. The results are reported in Figure 6, demonstrating that our proposed DGMamba can still capture the semantically related information, *i.e.*, the global shape and the object itself, when confronted with hard samples exhibiting significant disparity from the real world. Taking the 'dog' for instance, our proposed DGMamba is able to focus on the entire dog face, while the baseline VMamba suffers from recognizing these critical features. In addition, VMamba can be easily interfered by the background and the texture details, exampled by the 'house' and 'person'. While our DGMamba can still comprehensively learn the object information even with such complex domain-specific information. These findings highlight the excellence of our method in improving model generalization capacity.

## 5 CONCLUSION

This work is the first endeavor to explore the generalizability of the SSM-based model (Mamba) in DG. We propose a novel framework named DGMamba, that contains two pivotal techniques. Firstly, we design a novel Hidden State Suppressing to alleviate the adverse effect of domain-specific information conveyed in hidden states. Secondly, we propose Semantic-aware Patch Refining, which consists of Prior-Free Scanning and Domain Context Interchange. Both aim to direct the model's focus toward the object rather than the context. Extensive experiments on four widely used DG benchmarks show the superiority of DGMamba compared with state-of-the-art DG methods based on CNN or ViT. We believe our work builds a solid baseline for exploiting SSMs for the DG community. In the future, we would like to investigate the feature prompt or domain prompt to facilitate SSM-based models learning more powerful representations for enhancing the model generalizability.

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
