# OpenReview forum: "DGMamba: Domain Generalization via Generalized State Space Model"
_acmmm.org/ACMMM/2024/Conference — MM2024 Poster_

### Official Review · Reviewer_WKDQ · 2024-05-25

**Rating:** 2
**Confidence:** 2

**Summary:**

This paper introduces Mamba to the task of DG. The proposed method consists of two modules HSS and SPR to address the issues when applying Mamba to DG. Ths paper is well written and the description of the proposed method is clear. The overall idea in this paper is generally novel (but not too much). However, there are more concerns to the paper from the problem motivation to the rationale behind the proposed modules.

**Strengths:**

1.	This paper is well written and easy to follow.
2.	How to apply the Mamba architecture to the task of DG is interesting.

**Limitations:**

1.	The motivation of this paper is not strong. Mamba is good, the task of DG is important, which doesn’t mean that we have to apply Mamba to DG. What are the advantages of applying Mamba to DG compared to ViT? Does the state space model have any specifically good property for DG? If not, applying Mamba to DG is simply a trivial “A+B” work for pursuing the hot topic. Is section 3, the authors discussed the shortcomings of Mamba for DG, but without describing the advantages of Mamber for DG. So why must we apply Mamba to DG? Is it only because Mamba is hot and XXXMamba is easy for publication?
2.	The authors claim that the domain-specific information is accumulated in hidden states, which should be suppressed. Can this speculation be validated, because it is an important motivation to design HHS? Besides, the design of HHS is somehow heuristic. What is the retionale behind simply using a threshold to suppress delta A?
3.	What is the reason to introduce of SPR with the existing of HSS? What problems of HHS are left behind for SPR to solve? In addition, handling the relationship between context and object seems not quite related to the architecture of Mamba. It can also be applied as a plug-in to ViT-based DG methods. Therefore, HHS and SPR seem to be indepent each other.
4.	As an important hyperparameter, does the selection of threshold alpha vary according to the dataset? The experiment shown in Figure 4 only gives the result on PACS.
5.	In the comparison of between HSM and HSS in Table 7, what are the results when alpha=0.3 and 0.35, because they are shown in Figure 4?
6.	Through all the experiments, i.e. SOTA comparison, ablations, feature visualization, activation map, it is still hard to make me convinced that HHS learn more domain-generic knowledge. Besides, the experiments cannot show how and why SPR works. More specifically designed experiments for model validation is needed. Some routine experiments (hyperparameter sensitivity, computational efficiency, feature visualization, and activation map) are not quite sufficient to validate the effectiveness of the proposed modules.

**Suitability:**

2

---

### Official Review · Reviewer_vghi · 2024-05-26

**Rating:** 3
**Confidence:** 4

**Summary:**

Based on Mamba, the authors propose DGMamba to enhance the model's domain generalization. It consists of two parts: Hidden State Suppressing(HSS) and Semantic-aware Patch Refining(SPR). SPR consists of two core modules: Prior-Free Scanning (PFS) and Domain Context Interchange (DCI).

**Strengths:**

1 HSS mitigates the influence of accumulated domain-specific information in hidden states.
2 SPR assists the model in better focusing on the object.
3 It achieves state-of-the-art generalization performance.

**Limitations:**

1 The reported results of GMoE are lower than the results publicly disclosed in its article.
2 The original Mamba has achieved the state-of-the-art generalization performance on the PACS dataset. DGMamba shows limited improvement. Has Mamba also achieved the best results on other datasets?
3 Is an additional model required for Grad-CAM? Is the Grad-CAM used only in training.
4 The article lacks results for the SVIRO, Wilds-Camelyon, Wilds-FMOW, and DomainNet datasets, which are also commonly used for evaluating domain generalization.

**Suitability:**

2

---

### Official Review · Reviewer_k334 · 2024-05-27

**Rating:** 4
**Confidence:** 3

**Summary:**

The paper introduces DGMamba, a novel framework for domain generalization (DG) that aims to address distribution shift problems in various scenes. Domain generalization is challenging for multimedia and computer vision systems because they struggle to recognize images with domain distribution shifts, such as changes in background and lighting conditions. Existing DG methods rely heavily on Convolutional Neural Networks (CNNs) or Vision Transformers (ViTs), which have limitations such as limited receptive fields or quadratic complexity issues. DGMamba leverages the advantages of a state space model (SSM) called Mamba, which offers superior linear complexity and global receptive fields. However, Mamba faces challenges with hidden state issues and inappropriate scan mechanisms when applied to DG. To overcome these, DGMamba incorporates two core components: Hidden State Suppressing (HSS) and Semantic-aware Patch Refining (SPR). Extensive experiments on four DG benchmarks demonstrate that DGMamba achieves superior results compared to state-of-the-art models.

**Strengths:**

The first work to introduce Mamba into the field of domain generalization (DG) was exceptionally comprehensive, meticulously covering both theoretical foundations and practical implementations. The experimental results presented by the author were particularly impressive, achieving state-of-the-art generalization performance across a variety of challenging datasets. These results not only demonstrated the efficacy of Mamba in enhancing generalization but also provided valuable insights into the underlying mechanisms that contribute to its success. By pushing the boundaries of what was previously thought possible, this work has opened up new avenues for research and application, offering a fresh and innovative perspective for tackling the complex challenges associated with domain generalization problems. Through rigorous experimentation and detailed analysis, the author has laid a solid groundwork for future studies, encouraging further exploration and refinement of Mamba-based approaches in the quest for more robust and adaptable machine learning models.

**Limitations:**

1.Why can Prior-Free Scanning make images have causal correlation? Lack of corresponding theoretical derivation.
2.Were the experimental results averaged over multiple runs? What protocol was used in the experiment? The experimental section does not clearly detail these.
3. As mentioned in Table 5, VMamba has an accuracy of 89.4%, which exceeds all other comparison methods in Table 1. How can the contradiction be explained between this and the statement in the paper "MambaOut: Do We Really Need Mamba for Vision?" that says, "We analyze common visual tasks against these criteria and argue that introducing Mamba for ImageNet image classification is unnecessary"?
4. Why is the accuracy of Naive Mamba (VMamba) on the PACS dataset mentioned as 87.7% on page 3, while Table 5 lists it as 89.4%?
5. The Grad-CAM visualization experiment lacks a comparison with the use of ResNet50 and DeiT-S models.
6. The code is not open source yet. Can it be easily reproduced after being open sourced?
7. The author can try EfficientVMamba-B, Vim-S, and other models to prove that the proposed module is effective in multiple visual Mamba models.

**Suitability:**

2

---

### Meta-Review · Area_Chair_QUNT · 2024-07-04

**Recommendation:** Accept (Poster)
**Confidence:** 3

**Metareview:**

- Summary of Strengths and Contributions

The paper introduces an approach to domain generalization (DG) by leveraging the Mamba state space model (SSM). The authors propose two core components, Hidden State Suppressing (HSS) and Semantic-aware Patch Refining (SPR), to address issues with hidden state accumulation and inappropriate scan mechanisms in Mamba when applied to DG. The paper is well-organized, clearly presented, and provides a comprehensive evaluation through extensive experiments on four DG benchmarks, demonstrating state-of-the-art performance.

- Summary of Weaknesses and Rebuttal Gaps

Despite the strengths, several critical weaknesses were raised by the reviewers:

1. Theoretical Derivation and Experimental Protocols (Reviewer k334): Reviewer k334 raised concerns about the lack of theoretical derivation for the Prior-Free Scanning mechanism and questioned the experimental protocols used. Additionally, there were inconsistencies in reported accuracies (e.g., Naive Mamba on the PACS dataset) and the need for Grad-CAM visualization comparisons. While the rebuttal provided partial clarifications, some issues remain unresolved.

2. Generalization and Dataset Coverage (Reviewer vghi): Reviewer vghi pointed out that the original Mamba already achieved strong performance, with DGMamba showing limited improvement on some datasets. The paper lacked results for several commonly used DG datasets like SVIRO, Wilds-Camelyon, Wilds-FMOW, and DomainNet. The authors’ response addressed some concerns but did not fully resolve the need for additional experiments and validations.

3. Motivation and Module Design (Reviewer WKDQ): Reviewer WKDQ questioned the motivation behind applying Mamba to DG and the rationale for the HSS and SPR modules. The reviewer also noted that the modules seemed independent and not fully integrated with the Mamba architecture. Concerns about hyperparameter sensitivity and the need for more convincing experiments to validate the modules’ effectiveness were partially addressed in the rebuttal.

4. Process Clarification: According to line 568, “DCI only performs on samples with high confidence in the classification results, while other samples remain unchanged.” This suggests that the proposed method requires forwarding samples first and then, based on the prediction confidence, forwarding them again. Please provide clarification on this.

- Final Recommendation

Despite the aforementioned weaknesses, the paper presents a fair contribution to the field of domain generalization. The introduction of DGMamba and the extensive experimental results demonstrating its efficacy. The authors have provided reasonable modifications to the Mamba framework, showing a good performance in domain generalization.

After considering the feedback from all reviewers and the authors’ responses, I recommend accepting the paper. Please make sure the camera-ready version includes all suggested changes, clarify the concerns raised by reviewers, and re-write the discussion on potential limitations as well as future directions. This recommendation has been discussed with SAC.